# Fatigue and Cognitive Dysfunction Are Associated with Occupational Status in Post-COVID Syndrome

**DOI:** 10.3390/ijerph192013368

**Published:** 2022-10-16

**Authors:** Cristina Delgado-Alonso, Constanza Cuevas, Silvia Oliver-Mas, María Díez-Cirarda, Alfonso Delgado-Álvarez, María José Gil-Moreno, Jorge Matías-Guiu, Jordi A. Matias-Guiu

**Affiliations:** Department of Neurology, Hospital Clínico San Carlos, San Carlos Health Research Institute (IdISCC), Universidad Complutense de Madrid, 28040 Madrid, Spain

**Keywords:** COVID-19, post-COVID syndrome, occupational status, fatigue, cognition, depression

## Abstract

Post-COVID syndrome (PCS) is a medical condition characterized by the persistence of a wide range of symptoms after acute infection by SARS-CoV-2. The work capacity consequences of this disorder have scarcely been studied. We aimed to analyze the factors associated with occupational status in patients with PCS. This cross-sectional study involved 77 patients with PCS on active work before SARS-CoV-2 infection. Patients were evaluated 20.71 ± 6.50 months after clinical onset. We conducted a survey on occupational activity and cognitive and clinical symptoms. The association between occupational activity and fatigue, depression, anxiety, sleep quality, and cognitive testing was analyzed. Thirty-eight (49.4%) patients were working, and thirty-nine (50.6%) patients were not. Of those not working at the moment of the assessment, 36 (92.3%) patients were on sick leave. In 63 patients (81.8% of the sample), sick leave was needed at some point due to PCS. The mean duration of sick leave was 12.07 ± 8.07 months. According to the patient’s perspective, the most disabling symptoms were cognitive complaints (46.8%) and fatigue (31.2%). Not working at the moment of the assessment was associated with higher levels of fatigue and lower cognitive performance in the Stroop test. No association was found between occupational status with depression and anxiety questionnaires. Our study found an influence of PCS on work capacity. Fatigue and cognitive issues were the most frequent symptoms associated with loss of work capacity.

## 1. Introduction

Post-COVID syndrome (PCS), or long-COVID, is a medical condition characterized by the persistence of a wide range of symptoms after the acute infection by SARS-CoV-2. According to the WHO definition, these symptoms should persist for at least three months after the acute onset and last for at least two months [1]. Initial estimates suggest a prevalence of up to 43%, depending on each symptom [2,3]. The most common symptoms reported by patients were fatigue, dyspnea, cognitive issues, anxiety, depression, insomnia, and pain [4,5]. PCS is possible at all ages, but it has mainly been linked to working ages and women [3]. Considering the range of symptoms and impairments, PCS can have severe consequences on the ability of patients to maintain their workforce. In this regard, reductions in work capacity, need for job adaptation, or inability to work may be important consequences. At the same time, work loss negatively impacts social participation, mood, and quality of life [6,7]. Due to the frequency of COVID-19 and PCS, this could impact the community due to long-term absences, loss of productivity, and invalidity. Thus, the influence of PCS on occupational outcomes represents a significant economic burden and public health. However, limited literature exists about the effects of PCS on workability [8,9,10].

PCS is a new disorder with heterogeneous symptoms and outcomes; therefore, it is essential to disentangle what symptoms or deficits are associated with greater consequences on the work status. This knowledge is relevant to guide the use and/or development of interventions to improve the most disabling symptoms. In addition, it is important to define the most significant clinical issues that should be evaluated to decide the capacity to work [11].

Thus, in this study, we aimed to evaluate the clinical characteristics associated with occupational outcomes in people with PCS. We included a consecutive cohort of patients with PCS that underwent a comprehensive clinical and neuropsychological examination. We evaluated the association between clinical characteristics and work status at the moment of the assessment and the need for sick leave due to PCS at any moment of the disease.

## 2. Materials and Methods

### 2.1. Study Design

We conducted a cross-sectional study including 82 patients evaluated with PCS between January and June 2022 in a specific program for PCS in the Department of Neurology of our center. Seventy-seven (93.9%) participants were employed before infection with COVID-19, two (2.4%) were unemployed, two (2.4%) were on sick leave, and one (1.2%) was off work due to a temporary disability. Thus, the analyses were restricted to the 77 patients previously on active work. The mean age of the patients was 46.31 ± 7.97 years old, and 67 (87.0%) were women. The time since the acute onset of COVID-19 and the assessment was 20.71 ± 6.50 months. The main demographic and clinical characteristics are depicted in Table 1.

Patients met the following inclusion criteria:-Diagnosis of COVID-19 confirmed by RT-PCR.-Diagnosis of PCS according to the WHO criteria [1].-Occupationally active before the onset of COVID-19.

Additionally, the following exclusion criteria were absent:-History of stroke, traumatic brain injury, or any neurological disorder before COVID-19.-Active psychiatric disorder not explained by PCS.-History of alcohol abuse or other toxics.-Any medical condition previously linked to reduced work capacity.-Sensory disorders, drugs or any medical or neurological condition potentially biasing the assessments.

The study was conducted with the approval of the local Ethics Committee, and all participants signed the informed consent.

### 2.2. Neuropsychological Assessment

Patients were evaluated with a comprehensive clinical, neuropsychological, and neuropsychiatric protocol, which has been specified elsewhere [12]. In brief, the patients were examined by a trained neuropsychologist using the following cognitive tests: Digit Span, Corsi’s Test, Symbol Digit Modalities Test (SDMT), Boston Naming Test (BNT), Rey–Osterrieth Complex Figure (ROCF), Judgement Line Orientation test (JLO), Stroop Color-Word Interference Test, Free and Cued Selective Reminding Test (FCSRT), verbal fluency (semantic and letter fluency) and the Visual Object and Space Perception Battery (VOSP). Furthermore, the State-Trait Anxiety Inventory (STAI) [13], Beck Depression Inventory-II [14], Pittsburgh Sleep Quality Index (PSQI) [15], Modified Fatigue Impact Scale [16], and Brief Smell Identification Test (BSIT) [17] were administered to evaluate anxiety, depressive symptoms, sleep quality, fatigue, and olfactory function. Patients were examined in a single session lasting approximately 1.5 h. 

### 2.3. Work Assessment

Patients were invited to complete a survey about their work situation. This survey is shown in the Appendix A and includes the following sections: work situation before the onset of the disease; need and duration of sick leave due to symptoms associated with PCS; job adaptations or changes; symptoms linked to work capacity according to the patient’s perspective. We used the ESCO framework (European Skills, Competences, Qualifications and Occupations) to classify professional categories [18]. Occupation is one of the three pillars of ESCO. We used the International Standard Classification of Occupations (ISCO 08) to categorize the occupations. According to the National Statistics Institute of our country [19], the distribution of workers during the first trimester of 2022 according to this categorization was as follows: Group 0: “Armed forces occupations” (men 1% and women < 1%); Group 1: “Managers professionals” (7.8% of men and 4.5% of women); Group 2: “Professionals” (21.79% of men and 29.84% of women), Group 3: “Technicians and associate professionals” (16.28% of men and 11.85% of women); Group 4: “Clerical support workers” (6.79% of men and 16.88% of women); Group 5: “Service and sales workers” (16.16% of men and 19.76% of women); Group 6: “Skilled agricultural, forestry and fishery workers” (men and women < 1%); Group 7: “Craft and related trades workers” (men 13% and women < 1%); Group 8: “Plant and machine operators and assemblers” (men 8% and women 1%); Group 9: “Elementary occupations” (men 6.7% and women 14%) [20]. The categorization takes into account the skill levels (i.e., the complexity and range of tasks and duties performed in an occupation).

### 2.4. Statistical Analyses

Data were analyzed using SPSS (IBM, version 26, Armonk, NY, USA). Descriptive statistics were used to summarize the data, using mean ± standard deviation or frequency (percentage). Kolmogorov–Smirnov tests were used to evaluate the normality of the distributions. Chi-squared tests were used to examine the differences between categorical variables. Mann–Whitney U tests were used to compare means between two groups. For all analyses, statistical significance was set at *p*-value < 0.05.

## 3. Results

### 3.1. Employment Status

Thirty-eight (49.4%) patients were working, and thirty-nine (50.6%) were not. Of those not working at the moment of the assessment, 36 (92.3%) were on sick leave. In 63 participants (81.8% of the sample), sick leave was needed at some point due to PCS. The repercussion on working capacity was general across the different occupational groups (Appendix A). 

The mean duration of sick leave was 12.07 ± 8.07 months. In patients on sick leave at the moment of assessment, the mean duration of sick leave was 14.80 ± 7.16 months.

### 3.2. Job Changes and Adaptations

Of the 38 patients who returned to work, 12 (31.6%) reduced their working hours. In addition, 18 (23.4%) needed some kind of job adaptation due to PCS, such as more breaks (6; 7.8%), telework (9; 11.7%), cognitive aids (4; 10.5%), or a position change (6; 7.8%). In addition, 10 cases reported the need to make adaptations but did not have that possibility. 

### 3.3. PCS Symptoms and Work Function

Seventy-five patients (97.4%) reported that PCS symptoms influenced their work capacity and professional career. Among the different symptoms interfering in work capacity, cognitive issues were regarded in 71 (92.2%), followed by fatigue (65; 84.4%), headache (59; 76.6%), sleep disorders (54; 70.1%), weakness (51; 66.2%), anxiety/depression (42; 54.5%), dizziness (43; 55.8%), dyspnea (46; 59.7%). Other symptoms reported by patients included palpitations (10; 13.0%) and gastrointestinal disorders (6; 7.8%). The most disabling symptoms, according to the patient’s perspective, were cognitive complaints (36; 46.8%) and fatigue (24; 31.2%). Other symptoms (dyspnea, headache, etc.) were regarded as the most disabling by fewer than 5% of patients. 

### 3.4. Association between Work Status and Clinical Factors and Assessments

The group that was not able to return to work was associated with higher levels of fatigue and worst cognitive performance in the Stroop test. There were no statistically significant associations with anxiety, depression, sleep quality and olfactory function. The need for ventilatory assistance during the acute disease was associated with a lower probability of returning to work, although only 10.3% of patients who were unable to return to work required ventilatory assistance during the acute phase and it was not statistically significant. Complete results are shown in Table 2 and Appendix A.

### 3.5. Association between Need of Sick Leave and Clinical Factors and Assessments

The group that required sick leave at some time of the disease showed higher levels of fatigue and depressive symptoms. In addition, these patients showed the worst cognitive performance on the Stroop test trial 2 (Color), Corsi test backward, and FCSRT (Free Recall Trial 1 and Total Recall) (Appendix A). These findings were also confirmed in the group of 73 patients who did not require ventilatory assistance or UCI admission (Appendix A). 

## 4. Discussion

In this study, we examined relevant factors associated with the working status of patients with PCS. We enrolled a cohort of patients with a predominance of women with high educational levels and jobs generally with intellectual demands. One of the most important findings is that many patients could not return to work 1–2 years after the symptom onset. In addition, 81.8% of the sample required sick leave due to PCS at some point, generally comprising several months. Furthermore, 31% reduced their working hours when returning to work, and 23.4% needed some adaptation. Overall, these results suggest an influence of PCS on working status and work capacity, according to recent evidence [21]. 

After a mean follow-up of 20.71 ± 6.50 months, half of the PCS patients at working age were not able to return to work. The inability to work at working ages has been demonstrated to be one of the main factors related to quality-of-life loss. Work loss due to the medical situation has been related to poorer quality of life [6] and resume work has been related to a better emotional status [7]. 

Another interesting finding of our study is the association between PCS and work capacity. We included the patient’s perspective using a questionnaire and analyzed the association between work status and scales and cognitive testing. The results were consistent between both approaches, confirming fatigue and cognitive function as the most critical factors associated with work capacity. Accordingly, this suggests that fatigue and cognitive function should be the main part of the assessment of work capacity and disabilities in PCS. However, the questionnaire responses revealed a wide variety of symptoms potentially interfering with work capacity, and a careful, individualized assessment of each symptom would be needed in these patients. Regarding cognitive function, the Stroop test was the most associated test, which is in accordance with previous studies in which the Stroop test is one of the most impaired tests in PCS and more associated with fatigue [12,22,23,24]. The results remained significant even after excluding those patients requiring ICU admission or ventilatory assistance. Conversely, psychiatric symptoms (anxiety levels and depressive symptoms in the context of PCS) were not associated with work status, and only depression was associated with the need for sick leave at any moment of the disease. This suggests that depression and anxiety are not the main symptoms linked to work capacity among the several symptoms present in the context of PCS. 

These results highlight the impact of SARS-CoV-2 infection on work status and emphasize the need for guidance, and eventually job adaptations (progressive return, adjustments in workload and tasks, etc.) during the process of recovery to successfully return to work [25]. Adjustments such as promoting telework when possible and reducing or fractionating workload could be useful to reduce fatigue and the impact of attentional difficulties. In addition, medical education of both patients and professionals could be important to adequately support patients with PCS, as has been suggested in patients with chronic fatigue syndrome [26,27]. Moreover, intervention and rehabilitation programs to relieve fatigue and improve cognitive performance may indirectly help in the work capacity and status of patients. At the same time, including outcomes related to work capacity may be convenient in clinical trials and studies evaluating treatments, rehabilitation programs and other interventions in the PCS [28]. 

Overall, these findings emphasize the potential long-term consequences of COVID-19 [29,30,31]. Our cohort is likely to be representative of patients attended in a specialized setting of patients with PCS. Further studies are needed to determine the incidence and prevalence, risk factors, and clinical course of PCS to evaluate the social, occupational, and economical consequences, especially from a population-based perspective due to the extension of SARS-CoV2 infection in most countries [32]. Regarding the occupational categories, our sample was reasonably representative of our region, although there was an overrepresentation of Group 2. In this regard, Groups 1, 2, and 3 (representing 80% of the sample in our study) are included within ISCO skill level 4, which involve performance on tasks requiring complex problem-solving, making decisions and creativity based on extensive theoretical knowledge in specialized fields. Thus, the potential influence of specific works should be further examined in future studies. In addition, longitudinal studies evaluating the course of PCS and the occupational consequences are also needed due to the fluctuating course of this condition and the largely unknown outcome. 

Our study has some limitations that should be considered. We included patients who attended our clinical setting, which could be a selection bias towards patients with a higher incidence of cognitive/fatigue issues. Although demographic and clinical characteristics are similar to other long-COVID cohorts reported in other clinical settings [33], our findings regarding the prevalence of sick leave and inability to return to work cannot be generalized and further studies in other cohorts and settings are needed. Another notable limitation is that we did not evaluate the quality of life, which could be important to assess the interplay between symptoms of PCS, work capacity and health outcomes. 

## 5. Conclusions

Our study found an influence of PCS on work capacity, with over 50% being unable to return to work more than one year after symptom onset and more than 80% requiring at least one period of sick leave at some point during the disease. Fatigue and cognitive issues were the most frequent symptoms associated with loss of work capacity. Thus, these symptoms should be carefully assessed when evaluating and treating patients with PCS due to their impact on work capacity. Further studies are necessary to confirm these findings. 

## Figures and Tables

**Table 1 ijerph-19-13368-t001:** Main demographic and clinical characteristics.

Variables		Mean ± SD or *n* (%)
Age (years)		46.31 ± 7.97
Sex (women)		67 (87.0%)
Months since acute onset to assessment		20.71 ± 6.50
Years of education		16.14 ± 3.03 years
ISCO Classification of occupations	Group 1	3 (3.9%)
	Group 2	44 (57.1%)
	Group 3	15 (19.5%)
	Group 5	10 (13.0%)
	Group 8	1 (1.3%)
	Group 9	4 (5.2%)
Hypertension		12 (15.6%)
Diabetes mellitus		6 (7.8%)
Dyslipidemia		19 (24.7%)
Smoking Habit		6 (7.8%)
COVID Reinfection		23 (29.9%)
Hospital Admission		15 (19.5%)
ICU Admission		3 (3.9%)
Ventilatory assistance		4 (5.2%)

SD: Standard deviation.

**Table 2 ijerph-19-13368-t002:** Comparison of patients on active working or not at the moment of the assessment.

	Returned to Work (*n* = 38)	Not Returned to Work (*n* = 39)	U/χ^2^	*p*-Value
*Demographic factors*				
Age	45.68 ± 8.58	46.92 ± 7.38	U = 409.50	0.475
Sex (women)	34 (89.5%)	33 (84.6%)	χ^2^ = 0.40	0.737
Education (years)	15.66 ± 3.14	16.62 ± 2.88	U = 612.50	0.110
Time since onset	21.84 ± 6.61	19.61 ± 6.28	U = 571.50	0.083
*Risk factors and clinical characteristics of acute disease*				
Arterial hypertension	6 (15.8%)	6 (15.4%)	χ^2^ = 0.002	0.961
Diabetes mellitus	4 (10.5%)	2 (5.1%)	χ^2^ = 0.78	0.431
Dyslipidemia	10 (26.3%)	9 (23.1%)	χ^2^ = 0.10	0.742
Hospital admission	5 (13.2%)	10 (25.6%)	χ^2^ = 1.91	0.167
Ventilatory assistance	0 (0%)	4 (10.3%)	χ^2^ = 4.11	0.115
ICU admission	0 (0%)	3 (7.7%)	χ^2^ = 3.04	0.24
*Fatigue and neuropsychiatric scales*				
MFIS (total)	55.24 ± 15.19	67.38 ± 9.89	U = 372.50	**<0.001**
MFIS (physical)	25.24 ± 7.042	30.28 ± 4.81	U = 390.50	**<0.001**
MFIS (cognitive)	24.68 ± 8.90	30.56 ± 5.34	U = 443.50	**0.002**
MFIS (psychosocial)	5.00 ± 2.29	6.36 ± 1.70	U = 470.50	**0.005**
BDI	14.65 ± 8.62	17.15 ± 7.32	U = 546.00	0.068
STAI-State	40.05 ± 9.54	42.62 ± 11.02	U = 655.50	0.383
STAI-Trait	47.08 ± 12.01	49.97 ± 12.54	U = 653.00	0.370
PSQI	10.61 ± 3.94	12.10 ± 4.29	U = 539.50	0.132
BSIT	9.53 ± 2.24	9.45 ± 2.25	U = 669.00	0.869
*Cognitive testing*				
Digit span forward	5.79 ± 1.37	5.62 ± 1.51	U = 675.50	0.494
Digit span backward	4.05 ± 1.22	4.36 ± 1.26	U = 629.50	0.239
Corsi forward	5.87 ± 1.095	5.56 ± 1.27	U = 646.00	0.315
Corsi backward	5.11 ± 1.26	4.87 ± 1.031	U = 671.00	0.46
SDMT	44.42 ± 13.73	39.74 ± 14.85	U = 598.00	0.145
Boston Naming Test	53.42 ± 4.62	53.64 ± 4.55	U = 717.50	0.81
ROCF copy(accuracy)	33.92 ± 2.83	34.21 ± 1.98	U = 723.50	0.853
ROCF copy (time)	132.63 ± 65.33	122.46 ± 42.61	U = 665.00	0.439
ROCF 3 min	20.03 ± 6.11	22.00 ± 6.41	U = 611.00	0.185
ROCF 30 min	19.90 ± 6.04	20.67 ± 6.122	U = 677.00	0.514
ROCF recognition	19.16 ± 2.47	19.51 ± 3.26	U = 709.50	0.746
Stroop W	95.50 ± 22.58	83.31 ± 26.17	U = 552.50	0.055
Stroop C	66.84 ± 15.75	57.62 ± 16.944	U = 523.50	**0.027**
Stroop W-C	42.00 ± 13.42	33.62 ± 11.60	U = 488.50	**0.010**
FCSRT (Free recall Trial 1)	7.97 ± 2.08	7.95 ± 2.27	U = 680.00	0.529
FCSRT (Total free recall)	28.37 ± 7.46	27.92 ± 6.14	U = 692.00	0.617
FCSRT (Total recall)	42.18 ± 7.18	41.44 ± 5.48	U = 616.50	0.203
FCSRT (Delayed free recall)	10.29 ± 3.01	10.08 ± 3.081	U = 730.50	0.914
FCSRT (Delayed total recall)	14.50 ± 2.68	14.21 ± 2.02	U = 610.50	0.159
Verbal fluency “Animals”	22.08 ± 6.39	21.79 ± 6.64	U = 676.00	0.507
Verbal fluency “P”	15.84 ± 4.32	17.10 ± 5.02	U = 624.50	0.234
VOSP Object decision	16.82 ± 2.27	16.62 ± 2.02	U = 676.50	0.506
VOSP Progressive silhouettes	8.39 ± 2.95	8.15 ± 1.98	U = 731.50	0.922
VOSP Position discrimination	19.29 ± 1.91	19.21 ± 1.88	U = 720.00	0.785
VOSP Number location	9.00 ± 1.69	9.00 ± 1.29	U = 719.50	0.816
JLO	23.45 ± 5.59	23.41 ± 4.69	U = 709.00	0.744

MFIS = Modified Fatigue Impact Scale; BDI = Beck Depression Inventory-II; STAI = State-Trait Anxiety Inventory; PSQI = Pittsburgh Sleep Quality Index; BSIT = Brief Smell Identification Test; SDMT = Symbol Digit Modality Test; ROCF = Rey–Osterrieth Complex Figure; Stroop W = Stroop Words; Stroop C = Stroop Color; Stroop W-C = Stroop Word-Color; FCSRT = Free and Cued Selective Reminding Test; VOSP = Visual Object and Space Perception Battery; JLO = Judgement Line Orientation test. Statistically significant *p*-values are shown in bold.

## Data Availability

The data presented in this study are available on request from the corresponding author.

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
