# Peer review of "Fatigue and Cognitive Dysfunction Are Associated with Occupational Status in Post-COVID Syndrome"

_ijerph, 2022, doi:10.3390/ijerph192013368_

Round 1
Reviewer 1 Report
In this study, the authors examine the association between LC fatigue and cognitive dysfunction and occupational status. The impact on capacitance in daily work routine has not been well studied. A cross-sectional approach was chosen for the study, like most studies in this field. The authors' effort in recruiting participants, the extensive testing, and the categorisation of work status has to be recognised. The cohort was well defined and chosen with reasonable exclusion criteria covering possible confounding conditions. Nonetheless, there is a bias in the clinical setting, a sound but low-key statistical analysis with sometimes overinterpretation and -generalisation of the results.
For the results, the most common complaints were fatigue and cognitive impairments, as shown in most other PCS-cohorts. The conclusion that those complaints are most strongly associated with workability is obvious, the most complaint symptoms.
Interestingly, the mean sick leave duration was a year on average, which seems a pretty long time. Nearly all patients requiring sick leave have comorbidities they complain of. So, this might be an influencing factor that must be considered when drawing conclusions from the data set.
In the cohort, most participants did fall into category 2 of work status. Hardly anyone does work in groups 1, 8 and 9. Presumably, categories 8 and 9 are not common in a developed country. That only 3 participants working in group 1 are presenting themselves in the centre is worth mentioning and should make us think about why this is the case. Sure enough, there might be a bias in recruiting participants. Those who suffer most do present themselves in the department. Why are groups 2 and 3 overrepresented in these recruitments? I admit that it is above the scope of this study, but these should be the questions addressed by research in this field. Where are the ones who are self-employed or work in responsible/executive positions? Authors mentioned in the discussion the “intellectual demands” for the occupancy.
Minor points:
- It is well known that those heavily affected in the initial infectious state have stronger symptoms in the post-infectious phase. What does the comparison between the different groups (working/not working or sick-leave/ no sick-leave) look like when those seven participants who received ventilatory assistance or ICU admission are ruled out?
- The conclusion that psychiatric co-morbidities were not associated with work status or otherwise is unclear. Are these co-morbidities induced by COVID, or are these co-morbidities not associated with COVID? Please clarify and adapt the conclusion.
- With the comment above on work categories, stratification for comparison is hardly possible with low representation of working groups 1, 8, and 9. Could authors please present critical findings of demographics, questionnaires, and nominal groups (working/not-working, sick-leave/ no-sick leave) for the different working groups?
- Are there factors impacting work capability (Sex, Age, etc.) that could be identified with multiple regression?
- The authors should emphasise that no general conclusion could be drawn from this data.
Authors only reinforce the common observation that LC-Complains affect workability and present a significant economic burden. I agree with the authors that there is an urgent need to address the impact on the economy and public health but is stating the obvious the key to finding a solution? LC complaints such as fatigue and low-cognitive performance are influenced on many levels. It does not seem simple, and this study does not try to disentangle the mutual influences of occupational status, fatigue, and cognitive complaints. There is no discussion about the underlying (psychological)-mechanism that might play a role. No concrete suggestions are made regarding how patients with PCS could be supported or treated.
I am sorry to say that yet another opportunity has been lost to disentangle the intricate relations of occupancy status with PCS in a more general way.
Author Response
Comments and Suggestions for Authors
In this study, the authors examine the association between LC fatigue and cognitive dysfunction and occupational status. The impact on capacitance in daily work routine has not been well studied. A cross-sectional approach was chosen for the study, like most studies in this field. The authors' effort in recruiting participants, the extensive testing, and the categorisation of work status has to be recognised. The cohort was well defined and chosen with reasonable exclusion criteria covering possible confounding conditions. Nonetheless, there is a bias in the clinical setting, a sound but low-key statistical analysis with sometimes overinterpretation and -generalisation of the results.
RESPONSE: Thanks for the positive comments about our article and for suggesting several interesting modifications. We have modified the Discussion (especially the Limitations section) and Conclusion section to reduce the overinterpretation and generalization of the results.
For the results, the most common complaints were fatigue and cognitive impairments, as shown in most other PCS-cohorts. The conclusion that those complaints are most strongly associated with workability is obvious, the most complaint symptoms.
Interestingly, the mean sick leave duration was a year on average, which seems a pretty long time. Nearly all patients requiring sick leave have comorbidities they complain of. So, this might be an influencing factor that must be considered when drawing conclusions from the data set.
RESPONSE: Thanks for this comment. In our study, we did not ask the patients what symptoms were present but what symptoms they considered impacted more in their job capacity. Thus, the patients pointed out fatigue and cognitive issues as the most relevant symptoms. These subjective symptoms are consistent with objective examination findings, especially about cognitive dysfunction. In our cohort, the prevalence of comorbidities was very low due to the strict inclusion and exclusion criteria that excluded other potential disorders that could bias assessments. Thus, sick leave was due to new-onset symptoms associated with long-covid, as shown in the questionnaire. We have modified the Methods and Discussion section to clarify these aspects.
In the cohort, most participants did fall into category 2 of work status. Hardly anyone does work in groups 1, 8 and 9. Presumably, categories 8 and 9 are not common in a developed country. That only 3 participants working in group 1 are presenting themselves in the centre is worth mentioning and should make us think about why this is the case. Sure enough, there might be a bias in recruiting participants. Those who suffer most do present themselves in the department. Why are groups 2 and 3 overrepresented in these recruitments? I admit that it is above the scope of this study, but these should be the questions addressed by research in this field. Where are the ones who are self-employed or work in responsible/executive positions? Authors mentioned in the discussion the “intellectual demands” for the occupancy.
RESPONSE: Thanks for this comment. In our cohort, groups 2 (Group 2: Professionals), 3 (Group 3: Technicians and Associate Professionals) and 5 (Group 4: Service and sales workers) are the most frequent in the study, followed by groups 9 (Group 9: “Elementary occupations), 1 (Group 1: Managers) and 8 (Group 8: “Plant and machine operators and assemblers”). The distribution in our study is quite similar to the data reported by the National Institute of Statistics of our country in our specific region and at a country level. In this regard, Groups 2, 5, 3, 4 and 9 are the most prevalent, both during the first and second trimesters of the year 2022. We have added a commentary in the Methods (showing the percentage of each occupation in the 1st trimester of 2022 in our region distributed by sex) and Discussion section. The percentage of Group 1 in our setting is very close to our sample; conversely, in our sample, there is an over-representation of Group 2, which we have included and discussed in the Limitations section of the revised version.
Regarding the authors' mention of "intellectual demands", the ISCO classification defines 4 skill levels to classify groups of occupations. “Skill level” is a function of the complexity and range of tasks and duties to be performed in an occupation. Skill level is measured operationally by considering one or more of the following aspects: the nature of the work performed in an occupation concerning the characteristic tasks and duties, the level of formal education and the amount of informal on-the-job training and/or previous experience in a related occupation required for the competent performance of these tasks and functions. In this regard, ISCO level 4 means the performance of tasks requiring complex problem-solving, decision-making, and creativity based on an extensive body of theoretical knowledge in a specialized field). Groups 2 and 3 (and also 1) are found within level 4, which confirms that the most represented works in our study involve intellectual demands.
Minor points:
- It is well known that those heavily affected in the initial infectious state have stronger symptoms in the post-infectious phase. What does the comparison between the different groups (working/not working or sick-leave/ no sick-leave) look like when those seven participants who received ventilatory assistance or ICU admission are ruled out?
RESPONSE: Thanks for this suggestion. We have added the analysis suggested by the reviewer. We have included two new tables in Supplementary Material, excluding those patients (Supplementary Tables 2 and 4). The findings remained significant even after excluding those patients requiring ICU or ventilatory assistance.
- The conclusion that psychiatric co-morbidities were not associated with work status or otherwise is unclear. Are these co-morbidities induced by COVID, or are these co-morbidities not associated with COVID? Please clarify and adapt the conclusion.
RESPONSE: We have clarified the Discussion as suggested by the reviewer. We meant psychiatric comorbidities associated with COVID, because other psychiatric disorders not explained by PCS were specifically excluded.
- With the comment above on work categories, stratification for comparison is hardly possible with low representation of working groups 1, 8, and 9. Could authors please present critical findings of demographics, questionnaires, and nominal groups (working/not-working, sick-leave/ no-sick leave) for the different working groups?
RESPONSE: Many thanks for this interesting suggestion. We have added this information to the manuscript following the reviewer’s suggestion. We have included a new table in Supplementary Material showing these data (please see Supplementary Table 1). We have reported only the findings for Groups 2, 3 and 5, because of the small sample size of the other groups.
- Are there factors impacting work capability (Sex, Age, etc.) that could be identified with multiple regression?
RESPONSE: Thanks for this suggestion. However, sex and age were not associated with work capability. Because most patients are within a relatively restricted age range and most are women, it is difficult to find differences in these demographic factors. Similarly, the work category does not seem to influence the work capability, at least in groups 2, 3, and 5.
- The authors should emphasise that no general conclusion could be drawn from this data.
RESPONSE: We have completed the Conclusions section emphasizing that further studies are necessary to confirm these findings. In addition, we have added that our results cannot be generalized in the Limitations section, and further studies in other cohorts and settings are needed.
Authors only reinforce the common observation that LC-Complains affect workability and present a significant economic burden. I agree with the authors that there is an urgent need to address the impact on the economy and public health but is stating the obvious the key to finding a solution? LC complaints such as fatigue and low-cognitive performance are influenced on many levels. It does not seem simple, and this study does not try to disentangle the mutual influences of occupational status, fatigue, and cognitive complaints. There is no discussion about the underlying (psychological)-mechanism that might play a role. No concrete suggestions are made regarding how patients with PCS could be supported or treated.
RESPONSE: Thanks for this comment. Although the impact of LC on workability is well-known in LC clinics, as the reviewer stated in the first comment, it has not been well studied. In this regard, finding the most significant symptoms impacting workability is a novelty of our study. In addition, we propose what cognitive tests and scales would be more associated with workability, which could be important to evaluate the work capacity of these patients. We agree with the reviewer that our study is largely descriptive, and no specific recommendations are made to how to treat these patients. The identification of the most important symptoms associated with work capacity is also important to prioritize treatments when available. We believe that, with this work, we have made interesting contributions to the field, although the current manuscript is only a first step.
I am sorry to say that yet another opportunity has been lost to disentangle the intricate relations of occupancy status with PCS in a more general way.
RESPONSE: Please see our response to the reviewer’s previous comments. We believe that our manuscript has been improved with all the reviewers’ suggestions, and we appreciate the opportunity to submit a revised version of our study. We are open to any other suggestions to improve our manuscript.
Reviewer 2 Report
Thank you very much for allowing me to read this interesting manuscript. The manuscript is based on a topic of great interest due to the impact of post-COVID syndrome with occupational status and could contribute to the knowledge about this. I appreciate the interest of researchers in trying to investigate the main characteristics of patients with PCS and how they influence their occupational status. Nevertheless, there are some comments and recommendations that I would like to make:
· The introduction is very interesting, it refers to quality of life, however, among the many tests chosen to assess the patient's condition no quality of life scale was used. Considering that PCS affects quality of life, why was none included in the study design?
· The methodology is very poorly described. Why non-parametric tests have been used, what is the purpose of each statistical test.
· Patients were assessed with a comprehensive clinical, neuropsychological, and neuropsychiatric protocol which has been specified in reference number 12. However, there are differences from the protocol in the original article, what is the purpose of the changes?
· What was the neuropsychological examination(duration, sessions, etc.)?
· Tables 1 and 2. It is not specified which results have been performed with Chi-squared test and U-Mann-Whitney tests.
· Results with a significant difference have not been highlighted in the tables. In such large tables it is easier to highlight this difference.
· The tables are very long, but they do not provide any particular information different from that included in the text, so it is worth considering whether this is the best way to present the results.
· Table 2 seems very repetitive with respect to the previous one and does not provide decisive information.
· Table two is constructed with the differences between two groups (Required sick leave (n=64)Did not require sick leave (n=14)), the difference between the samples seems high, is this difference significant? It is possible that the results we are getting are due to the limited sample size. This fact should be pointed out in the limitations of the study.
· Lines 194-196. The importance of rehabilitation is explained, but there are no specific data in the manuscript if the sample selected receives or has received rehabilitation.
· Again, the first part of the discussion revolves around quality of life, but although we can infer that cognitive impairment and fatigue undoubtedly affect quality of life, the study design did not take this into account. It is useful to incorporate new literature that discusses the most persistent cognitive impairments in people with PCS.
· Columns in table 1 have no title.
Author Response
Thank you very much for allowing me to read this interesting manuscript. The manuscript is based on a topic of great interest due to the impact of post-COVID syndrome with occupational status and could contribute to the knowledge about this. I appreciate the interest of researchers in trying to investigate the main characteristics of patients with PCS and how they influence their occupational status. Nevertheless, there are some comments and recommendations that I would like to make:
RESPONSE: Many thanks for the positive comments on our study and the interesting suggestions to improve the manuscript and future research.
- The introduction is very interesting, it refers to quality of life, however, among the many tests chosen to assess the patient's condition no quality of life scale was used. Considering that PCS affects quality of life, why was none included in the study design?
RESPONSE: Thanks for this interesting suggestion. Unfortunately, quality of life was not assessed. We have included it in the Limitations section.
- The methodology is very poorly described. Why non-parametric tests have been used, what is the purpose of each statistical test
RESPONSE: We have clarified the Statistical analysis section as suggested. Many of the variables were non-normally distributed. For this reason, we used non-parametric tests.
- Patients were assessed with a comprehensive clinical, neuropsychological, and neuropsychiatric protocol which has been specified in reference number 12. However, there are differences from the protocol in the original article, what is the purpose of the changes?
RESPONSE: Thanks for this question. In the first study, we included two batteries: a standardized (paper and pencil) battery, including most traditional tests, and a computerized battery. The standardized battery has been largely validated, and normative data for our setting is available. Findings were similar between both batteries, and with the first battery, the assessment of the main cognitive functions was achieved. For this reason, in this study, we shortened the assessment, and the computerized battery was removed.
- What was the neuropsychological examination (duration, sessions, etc.)?
RESPONSE: We have specified it in the Methods section (2.2).
- Tables 1 and 2. It is not specified which results have been performed with Chi-squared test and U-Mann-Whitney tests.
RESPONSE: We have specified the statistical test in Tables 1 and 2 as suggested.
- Results with a significant difference have not been highlighted in the tables. In such large tables it is easier to highlight this difference.
RESPONSE: Thanks for this suggestion. We have shown in bold the statistically significant p-values.
- The tables are very long, but they do not provide any particular information different from that included in the text, so it is worth considering whether this is the best way to present the results.
RESPONSE: Following this suggestion, we have maintained the first table of the results (Table 2) and moved it to Supplementary Material Table 3.
- Table 2 seems very repetitive with respect to the previous one and does not provide decisive information.
RESPONSE: Following the reviewer’s suggestion, we have moved Table 3, which is very similar to the previous one, to the Supplementary Material.
- Table two is constructed with the differences between two groups (Required sick leave (n=64)Did not require sick leave (n=14)), the difference between the samples seems high, is this difference significant? It is possible that the results we are getting are due to the limited sample size. This fact should be pointed out in the limitations of the study.
-> RESPONSE: Thanks for this suggestion. We have included this comment as a limitation of the study.
- Lines 194-196. The importance of rehabilitation is explained, but there are no specific data in the manuscript if the sample selected receives or has received rehabilitation.
RESPONSE: Thanks for this interesting suggestion. In our sample, the participation in rehabilitation programs was very heterogeneous, and it is not possible to draw conclusions. However, following this suggestion, we have included a comment emphasizing the interest in including work capacity outcomes in clinical studies and clinical trials evaluating the different rehabilitation procedures in PCS.
- Again, the first part of the discussion revolves around quality of life, but although we can infer that cognitive impairment and fatigue undoubtedly affect quality of life, the study design did not take this into account. It is useful to incorporate new literature that discusses the most persistent cognitive impairments in people with PCS.
RESPONSE: Thanks for this suggestion. We have added the lack of a measure of the quality of life in the Limitations section. In addition, we have incorporated more recent references about cognitive impairment in people with PCS, following the reviewer’s suggestion.
- Columns in table 1 have no title.
RESPONSE: We have amended the Table 1.